# Endogenous miRNA-Based Innate-Immunity against SARS-CoV-2 Invasion of the Brain

**DOI:** 10.3390/ijms24043363

**Published:** 2023-02-08

**Authors:** Walter J. Lukiw, Aileen I. Pogue

**Affiliations:** 1LSU Neuroscience Center, Louisiana State University Health Science Center, New Orleans, LA 70112, USA; 2Alchem Biotech Research, Toronto, ON M5S 1A8, Canada; 3Department of Ophthalmology, LSU Health Science Center, New Orleans, LA 70112, USA; 4Department Neurology, Louisiana State University Health Science Center, New Orleans, LA 70112, USA

**Keywords:** Alzheimer’s disease (AD), circular RNA (circRNA), COVID-19, microRNA (miRNA), SARS-CoV-2, single-stranded RNA (ssRNA), superior temporal lobe neocortex (STLN)

## Abstract

The severe acute respiratory syndrome *Coronavirus*-2 (SARS-CoV-2), the causative agent of COVID-19, possesses an unusually large positive-sense, single-stranded viral RNA (ssvRNA) genome of about ~29,903 nucleotides (nt). In many respects, this ssvRNA resembles a very large, polycistronic messenger RNA (mRNA) possessing a 5′-methyl cap (m**^7^**GpppN), a 3′- and 5′-untranslated region (3′-UTR, 5′-UTR), and a poly-adenylated (poly-A+) tail. As such, the SARS-CoV-2 ssvRNA is susceptible to targeting by small non-coding RNA (sncRNA) and/or microRNA (miRNA), as well as neutralization and/or inhibition of its infectivity via the human body’s natural complement of about ~2650 miRNA species. Depending on host cell and tissue type, in silico analysis, RNA sequencing, and molecular-genetic investigations indicate that, remarkably, almost every single human miRNA has the potential to interact with the primary sequence of SARS-CoV-2 ssvRNA. Individual human variation in host miRNA abundance, speciation, and complexity among different human populations and additional variability in the cell and tissue distribution of the SARS-CoV-2 angiotensin converting enzyme-2 (ACE2) receptor (ACE2R) appear to further contribute to the molecular-genetic basis for the wide variation in individual host cell and tissue susceptibility to COVID-19 infection. In this paper, we review recently described aspects of the miRNA and ssvRNA ribonucleotide sequence structure in this highly evolved miRNA–ssvRNA recognition and signaling system and, for the first time, report the most abundant miRNAs in the control superior temporal lobe neocortex (STLN), an anatomical area involved in cognition and targeted by both SARS-CoV-2 invasion and Alzheimer’s disease (AD). We further evaluate important factors involving the neurotropic nature of SARS-CoV-2 and miRNAs and ACE2R distribution in the STLN that modulate significant functional deficits in the brain and CNS associated with SARS-CoV-2 infection and COVID-19’s long-term neurological effects.

## 1. Introduction

The single-stranded viral RNA (ssvRNA) known as the severe acute respiratory syndrome *Coronavirus* 2 (SARS-CoV-2) that causes COVID-19 can be effectively inactivated by a number of natural ribonucleic-acid-based host cell defenses [1,2,3,4,5]. One of the most important of these defenses includes the actions of a class of small non-coding RNAs (sncRNAs) known as microRNAs (miRNAs) [1,2,3,4,5,6,7,8,9,10]. Via base-pair complementarity, miRNAs are capable of specifically targeting ssvRNA sequences such as the SARS-CoV-2 3′-UTR, 5′-UTR, and other internal coding and non-coding regions, thus promoting SARS-CoV-2 inactivation and/or neutralization [5,6,8,9,10,11,12,13,14,15]. The essential molecular-genetic mechanism is very much similar to RNA interference (RNAi), a biological process in which natural RNAs are involved in sequence-specific suppression of gene expression by double-stranded RNA through transcriptional or translational repression [14]. RNA-sequencing, in silico and bioinformatics analysis, and molecular-genetic investigation indicate that most naturally occurring human miRNAs have extensive complementarity to the SARS-CoV-2 ssvRNA genome [2,3,15,16,17]. As miRNA abundance, speciation, and complexity vary significantly among human individuals, this may in part explain the variability in the innate-immune and pathophysiological response of different individuals to SARS-CoV-2 invasion and overall susceptibility to ssvRNA-mediated viral infection and the development and severity of COVID-19 infection [14,15,16,17,18,19]. For the first time, we report the 54 most abundant miRNAs in the human superior temporal lobe neocortex (STLN), an evolutionarily recent anatomical region of the brain involved in processing auditory information, cognition, memory, and multiple aspects of visual perception [15,18,19,20,21,22]. The high incidence of SARS-CoV-2 interaction with the STLN may be due in part to the relatively high density of the most important SARS-CoV-2 cell surface receptor, the angiotensin converting enzyme-2 (ACE2) receptor (ACE2R) in the STLN [19,20,21,22]. This may in part explain ‘brain fog’ and cognitive deficits associated with COVID-19 infection, especially in the more severe cases of COVID-19 infection in the elderly, as well as the long-term COVID-19 recovery period (long COVID) from SARS-CoV-2-induced neurological damage [20,21,22]. In the current paper, **(i)** we review some recently described aspects of miRNA and ssvRNA ribonucleotide sequence structure in this highly evolved miRNA-ssvRNA recognition and signalling system; **(ii)** for the first time, we report the most abundant miRNAs in control of the superior temporal lobe neocortex (STLN), an anatomical area of the association cortex of the human brain involved in cognition and targeted by SARS-CoV-2 invasion and Alzheimer’s disease (AD); **(iii)** we review which of these miRNAs are known to be induced by the pro-inflammatory transcription factor NF-kB (p50/p60) and those STLN-enriched miRNAs that are significantly induced in AD-affected brains; and **(iv)** we evaluate important factors involving the remarkable neurotropic nature of SARS-CoV-2 and the miRNA and ACE2R distributions in the STLN that modulate significant neurological deficits associated with successful SARS-CoV-2 invasion and COVID-19 infection.

## 2. Human Host Brain-Enriched microRNA (miRNA) and the SARS-CoV-2 ssvRNA

Initially characterized in both the plant and animal kingdoms just over ~2 decades ago, single-stranded non-coding RNAs (sncRNAs), known as microRNAs (miRNAs), **(i)** have since been detected and partially characterized in every single plant and animal species, tissue, and cell type so far analyzed [6,7,8,10,11]; **(ii)** have revealed themselves as essential post-transcriptional regulators of messenger RNA (mRNA) speciation, abundance, and complexity in multiple cell and tissue types [6,8,10]; **(iii)** have been shown to perform significant and determinant functions in plant and animal health, development, aging, and disease [11,12,13,14,15,16,17]; and **(iv)** have been implicated in basic molecular-genetic mechanisms in cardiovascular disease, cancer, Alzheimer’s disease (AD), prion disease (PrD), age-related neurodegeneration, and many other human systemic and neurological pathologies [18,19,20,21,22,23,24,25,26,27,28,29,30]. All mature miRNAs are derived from double-stranded RNA (dsRNA) precursors generated by RNA Pol II or RNA Pol III that, in most eukaryotes, are processed into smaller ~18–25 nt mature miRNAs by RNaseIII or RNaseIII-like nuclear processing enzymes [6,10,24,26]. Interestingly, **(i)** miRNAs are probably the most highly selected and information-dense of all sncRNA species [23,28,30]; **(ii)** the integrated use of chip-based miRNA arrays and microfluidics, RNA-sequencing technologies, applied bioinformatics, miRNA–mRNA analytical algorithms, and biostatistical studies indicate that a ~22 nucleotide (nt) sncRNA, the size of a typical eukaryotic miRNA, consisting of just four different ribonucleotides (adenine, A; cytosine, C; guanine, G; or uridine, U) has the potential to generate well over ~10**^12^** miRNA sequence combinations. Actual abundance-based studies, however, have indicated that only a total of about ~2650 miRNAs of different primary sequence are present in the entire human transcriptome. Perhaps even more remarkable is that there are only about ~50 different miRNAs in the human brain, central nervous system (CNS), and retina, suggesting an unusually high selection and evolutionary pressure to utilize only highly specific miRNA sequences to function in biologically useful miRNA–mRNA interactions [11,12,13,14,20,24,25,26,27,28,29,30,31]. This fascinating and complex miRNA–mRNA recognition system based on base-pair complementarity ultimately regulates transcriptional output and shapes patterns of global gene expression and the composition of the transcriptome in development, aging, and disease [11,12,13,14,15,16,17,18,23,24,25,26,27,28,29,30]. What is also remarkable is that only about ~2.65 × 10**^3^** miRNAs in 10**^12^** potential miRNA sequences have engendered useful molecular-genetic functions in miRNA–mRNA-based gene regulation and signaling in all of human biology [29,30,31]. Put another way, less than 1 in about ~2 × 10**^9^** potential miRNA sequences have found a useful purpose in the miRNA–mRNA-based gene regulation of global gene expression patterns in human biology, and many of these functions appear to involve innate-immunity and the protection of host cells and tissues from ssvRNA invasion [2,3,4,5,28,29,30,31].

Of additional interest is that there are recent reports that miRNA or anti-miRNA sequences may take the form of closed miRNA circles containing tandem miRNAs in a head-to-tail arrangement that, because of their lack of free 3′ or 5′ termini, are resistant to endo- and exo-nuclease cleavage, and hence are extremely stable and preserved in structure and function over long periods of time [32,33,34,35]. It has been recently shown that these circular miRNAs (circRNAs) may be especially important in the natural processes of neurogenesis, neurodevelopment, and neurodegeneration or in innate-immunity wherein miRNAs and/or their respective anti-miRNAs (AMs) have the potential to modulate host resistance to multiple types of ssvRNA, and hence their potential for infectivity [32,33,34,35].

## 3. Tissue and Cellular Variability in the Angiotensin-Converting Enzyme-2 (ACE2) Receptor (ACE2-R)

As many independent laboratories have shown, the major gateway for invasion of the SARS-CoV-2 virus into human host cells is via the angiotensin-converting enzyme 2 (ACE2) transmembrane receptor (ACE2R) expressed in multiple immune and non-immune human cell types [36,37,38,39,40,41], SARS-CoV-2 thus has the remarkable and unusual capacity to attack many different human host cells simultaneously via novel caveolae- and clathrin-independent endocytic pathways, becoming injurious to diverse cell, tissue, and organ systems, while exploiting any immune weakness in the host [38,39,40]. Further, co-localization and cross-linking studies have suggested that cholesterol- and sphingolipid-rich lipid raft micro-domains assist in the SARS-CoV-2–ACE2R interaction and subsequent SARS-CoV-2 entry into susceptible cells via the ubiquitous ACE2R [36,37,38,39,40,41]. The elicitation of this multipronged attack explains in part the severity and extensive variety of signs and symptoms observed in patients afflicted with COVID-19 [38,39,40,41]. One recent analysis of ACE2 expression in 85 human tissues including 21 different brain regions, 7 fetal tissues, and 8 receptor controls indicated that, besides strong ACE2 expression in the respiratory, gastrointestinal (GI), excretory, renal, reproductive, and immune cells, high ACE2 expression was also found in the amygdala, brainstem, and neocortical association regions within the cerebral cortex or neocortex [39,40,41]. Some of the highest ACE2 expression levels have been found: **(i)** in the pons and medulla oblongata of the human brainstem, containing the Botzinger complex, located in the rostral ventrolateral medulla and ventral respiratory column, that includes the primary respiratory control centers of the brain [40,41]; and **(ii)** in the limbic regions of the brain including the hippocampal formation (*cornu ammonis* CA1 and CA2 regions) and the temporal lobe neocortex [20,38,39,40,41]. High ACE2 expression in the medullary respiratory centers may in part explain the susceptibility of many COVID-19 patients to severe respiratory distress, while high ACE2R expression in the limbic system may in part explain lingering COVID-19 effects such as ‘brain fog’ and ‘long-term COVID’ with long-term neurological deficits [40,41]. Interestingly, the human hippocampus and superior temporal lobe neocortex (STLN) are enriched with many of the same miRNAs—perhaps not too surprising because these two major anatomical regions of the human limbic system and the paleomammalian cortex share common behavioral and signaling functions including audition; the processing and integration of auditory information; and multiple aspects of cognition, memory, and visual perception [15,18,19,20,21,22,41]. These anatomical regions are the very same ones targeted by the AD process and have long been known to exhibit the most significant AD-type inflammatory neuropathology [22,27,41].

## 4. Human Biochemical Individuality—Why Not All COVID-19 Patients Are Equally Affected

Almost 50 years ago, the two-time Nobel laureate, Linus Pauling, first described the significant biological variation in genotypic versus phenotypic parameters among human individuals, based in part on studies from hemoglobin genetics [42,43,44,45,46,47]. Overall, this concept has come to be known as “*human biochemical individuality*” [43,44,45,46]. This idea currently forms the basis for our evolving perception of “human genetic individuality” and our ongoing efforts to better understand the genotypic basis of human phenotypic diversity in development, health, senescence, and disease [43,44,45,46]. More recently, large population studies have analyzed the contribution of variability in gene expression, including the impact of genetic mutations, to “human genetic normalcy”, “human genetic individuality”, phenotype, susceptibility to disease, and related parameters that include the general redundancy in human gene expression patterns, which can directly impact the genetic evolution of the human species and overall susceptibility to disease. These patterns are currently thought to impact and in part define molecular-genetic functions of the human innate-immune system and host susceptibility to infectious disease, including infection by lethal ssvRNA viruses such as SARS-CoV-2 and neuroprotection by sncRNAs that include brain-enriched miRNAs [46,47,48,49,50,51] (Table 1). The widely observed variance in miRNA abundance, speciation, and complexity in normal, healthy aging in individuals defines in part what particular miRNA species are present, and specific patterns of miRNA abundance appear to have an impact on the particular physiological or pathophysiological status of individual susceptibility to both viral invasion and progressive inflammatory neurodegeneration [29,30,41]. Interestingly, a defined set of overexpressed miRNAs has been associated with an increased risk for the development of AD; therefore, it is reasonable to suggest that a specific set of host miRNAs and their specific cellular abundance may also confer susceptibility to, or protection against, ssRNA-based infectivity, viral-based infectious disease, and/or other forms of age-related inflammatory neurodegeneration [51,52,53,54,55,56,57,58,59,60,61,62,63,64,65,66,67].

## 5. Unanswered Questions

A large number of unanswered questions remain regarding endogenous host miRNAs and their potential contribution to human neurological immunity, including that against SARS-CoV-2-based ssvRNA-mediated viral invasion. These include the following: **(i)** miRNAs contain very highly selected sncRNAs, as is reflected in their primary ribonucleotide sequence; for example, as mentioned earlier, a 22 nt ssRNA with the possibility of four ribonucleotides at each position yields 10**^12^** possibilities—but the fact that only 2650 miRNAs are present indicates a very strong selection pressure with only 1 in about ~10**^9^** miRNA possible sequence combinations or ‘variants’ being present—and presumably utilized—in eukaryotic biology. The question that arises is that why were only these specific ~2650 miRNAs selected for use in human biology [11,12,13,14]?; **(ii)** Biological evolution designs are very elaborate and very often redundant systems exist for immune and neurological function and neuroprotection; was there some type of co-evolution of miRNA sequence and target ssvRNA sequences that might explain their remarkable affinity for each other, and has this outcome been advantageous to the host, to the invading ssvRNA species, or to both [12,14,15]?; **(iii)** The SARS-CoV-2 ssvRNA has the potential to interact with most of the ~2650 host miRNAs; however, it remains unclear what miRNAs are the most efficient in ‘deactivating’ the SARS-CoV-2 ssvRNA to alter its viability and prevent viral invasion and successful infection [15,16]; **(iv)** As for host miRNA sequences, can multiple miRNAs target multiple SARS-CoV-2 ssvRNA binding sites at the same time?; **(v)** Regarding miRNA and sncRNA propensity for targeting multiple mRNAs, are other regulatory, viral, and/or non-viral mRNAs targeted at the same time as part of a complex global miRNA- or sncRNA-based innate-immune regulatory system?; **(vi)** Do different host cell types and tissue systems utilize the same or different miRNA-based targeting and silencing mechanisms in the neutralization of ssRNA viruses such as SARS-CoV-2?; **(vii)** Besides miRNA and sncRNA, are other structural forms of RNA capable of binding to and neutralizing SARS-CoV-2 and/or other ssRNA viruses, and could this be used therapeutically in the clinical management of COVID-19 infection?; **(viii)** Circular RNAs (circRNAs), especially abundant in the brain association neocortex and retina, are known to contain both miRNA and natural anti-miRNA (AM; antagomir) sequences—do circRNAs possess specific anti-viral properties, do they participate in neuroprotective schemes, and could this novel form of ssRNA be exploited therapeutically to neutralize pathological SARS-CoV-2-mediated activities?; **(ix)** It is currently not clear if lingering or incomplete miRNA-SARS-CoV-2 binding may be associated with modulation or moderation of SARS-CoV-2 infectivity and contribute to such ancillary COVID-19 complications that include ‘brain fog’ or ‘long COVID’. This short list of ‘unanswered questions’ compiled by the authors are for the specific benefit of the readers and researchers as to what the focus of future research may be in this particular area of biomedical investigation. This list was in part provided to engender future questioning and research in this fascinating area of neural virology and neurobiology that may impact the onset and propagation of future viral infections and pandemics - and neuroinvasion- by a particular viral strain. A more complete understanding of these matters in the brain and CNS will result in improved anti-SARS-CoV-2 therapeutic approaches that will dampen the invasiveness and neuro-infectious potential of SARS-CoV-2 and the proliferation of COVID-19 in susceptible human populations.

## 6. Summary

The biological process of ssvRNA-mediated infection is a very complex biological undertaking by the invading viral species, the human cell-surface receptors, and the immune response of the host cell, and is also technically challenging to analyze and investigate, especially in potentially transmissible and lethal neurological disorders requiring a high level of biological containment. As time passes, we will continue to acquire more complete data on the SARS-CoV-2 infection mechanism, as well as the impact of ‘brain fog’, ‘long COVID’, and other neurological complications in larger human populations affected by COVID-19. Importantly, SARS-CoV-2 is highly neurotropic towards the human brain and CNS, a property in part based on the brain’s extensive neural vasculature and potential for viral particle delivery and the capability of this ssvRNA to invade the association neocortex, hippocampal formation, and related limbic regions of the human brain and CNS. These anatomical regions include the superior temporal lobe neocortex (STLN), known to be centrally involved in audition, memory, and cognition, which may in part explain the establishment of long-term neurological complications and a ‘lingering’ cognitive impairment. This may be especially important in susceptible COVID-19 patients, and especially in the elderly and people who have survived more serious episodes of COVID-19 infection [59,60,61,62,63,64,65,66]. Interestingly, for the limited research that has already been undertaken, certain natural miRNAs such as the NF-kB (p50/p65)-inducible hsa-miRNA-146a-5p are significantly induced in host cells in response to viral invasion, but whether this is part of a successful invasion strategy for the virus or part of a neuroprotective mechanism is still not well understood [17,18,19,51,52,53,54,55,56,57,58]. In fact, miRNA-146a-5p is significantly over-expressed in progressive and often lethal viral- and prion-mediated and related neurological syndromes associated with progressive inflammatory neurodegeneration, including at least ~18 different viral-induced encephalopathies, such as SARS-CoV-2, and for which data are currently available [16,41,49,50,51]. miRNA-146a is also significantly up-regulated in AD and PrD [2,16,57,58]. Interestingly, up to ~35% of all COVID-19 patients experience neurological and/or neuropsychiatric symptoms, and a pre-existing diagnosis of AD continues to predict the highest risk for COVID-19 yet identified, with the highest mortality among the most advanced cases and the most elderly of AD patients [5,62,63,64,65,66,67]. SARS-CoV-2 may potentiate the severity of neurological and neurocognitive deficits in patients already afflicted with other lethal neurodegenerative disorders such as AD and PrD [59,60,61,62,63,64,65,66,67]. Accordingly, the basal miRNA composition of the AD and/or COVID-19 patient and the prevention, diagnosis, and management of sustained neuropsychiatric manifestations of COVID-19 continue to encompass critical health care directives and provide a compelling rationale for the careful monitoring of COVID-19 patients across the entire infection spectrum. This includes early intervention and mitigation efforts that may reduce or modulate both the development of ‘brain fog’ and the longer-term neurological complications often associated with the multiple sequelae of post-COVID-19 disease.

## Figures and Tables

**Table 1 ijms-24-03363-t001:** Quality control of brain tissue pools and total RNA quality from the superior temporal lobe neocortex (STLN) used in these studies.

Age Range (years)	Mean Age ± 1 SD (years)	Gender	PMI	Total RNA Quality
65–78	75.2 ± 9.5	9 M, 18 F	1.3–3.3 h	1.9–2.1

Legend to Table 1: Age Range (years) = age range of control tissues used in these studies (in years); Mean Age = average age of all superior temporal lobe neocortical tissue samples (STLN; Brodmann Area 22) in years plus or minus one standard deviation (SD); PMI = post-mortem interval (death to brain freezing interval at −81 °C); range in hours; numbers in the ‘total RNA quality’ column are indices for RNA spectral purity and represent the ratio of spectral absorbance of RNA scanned at 260 nm/280 nm, based on a total N = 27; the use of RNase-free plastic-ware and extraction reagents containing RNAsecure (Ambion-Invitrogen; ThermoFisher Scientific Inc., Waltham, MA, USA) ribonuclease inhibitors and short PMI brain tissues of the highest qualities obtainable yielded high-spectral quality total RNA, as analyzed using RNA LabChip Analysis Chips (Caliper Life Sciences Inc.-PerkinElmer, Waltham MA, USA;) and a 2100 Bioanalyzer (Agilent Technologies; Santa Clara, CA, USA) [68,69]; miRNA fractions were obtained from total STLN RNA using selective small RNA extraction and an miRNeasy Micro Kit [50] (Cat.No./ID: 217084; Qiagen, Germantown, MD, USA) and were analyzed on miRNA analytical arrays (proprietary MRA-1001-miRNA microfluidic chip analytical platform (LC Sciences Corporation, Houston, TX, USA; https://lcsciences.com/company/technology/technology-microarray/ (accessed on 26 December 2022); data derived from [50,51,55,58,61,65,66,67,68,69]; see below and Table 2).

**Table 2 ijms-24-03363-t002:** Relative abundance and speciation of miRNA in the *Homo sapien* (hsa) brain superior temporal lobe neocortex (STLN; Brodmann Area 22); data derived from a pool of 27 control brains (see Table 1); [23,40,41,42,48].

hsa-let-7a **	hsa-miRNA-28	hsa-miRNA-132 *
hsa-let-7b *	hsa-miRNA-29a	hsa-miNA-143
hsa-let-7c **	hsa-miRNA-30b *^f^	hsa-miRNA-145 *
hsa-let-7d **	hsa-miRNA-30c	hsa-miRNA-146a *^f^
hsa-let-7e *	hsa-miRNA-30d *	hsa-miRNA-155 ^*f^
hsa-let-7f *	hsa-miRNA-34a *^f^	hsa-miRNA-181a *
hsa-let-7g **	hsa-miRNA-99a	hsa-miRNA-181b
hsa-let-7i **	hsa-miRNA-99b	hsa-miRNA-181d
hsa-miRNA-7 *	hsa-miRNA-100	hsa-miRNA-185
hsa-miRNA-9 *^f^	hsa-miRNA-103	hsa-miRNA-191 *
hsa-miRNA-16 *	hsa-miRNA-107 *	hsa-miRNA-195
hsa-miRNA-23a	hsa-miRNA-124a *	hsa-miRNA-221
hsa-miRNA-23b	hsa-miRNA-125a	hsa-miRNA-222 *
hsa-miRNA-24	hsa-miRNA-125b **^f^	hsa-miRNA-223
hsa-miRNA-26a	hsa-miRNA-126	hsa-miRNA-320 *
hsa-miRNA-26b	hsa-miRNA-127	hsa-miRNA-342 *
hsa-miRNA-27a	hsa-miRNA-128a *	hsa-miRNA-361
hsa-miRNA-27b *	hsa-miRNA-128b *	hsa-miRNA-485

Legend to Table 2: This table indicates the most abundant unique miRNAs detected in the human brain superior temporal lobe neocortex (STLN; Brodmann Area 22); abundances are ranked by miRNA numerical designation, based on pooled data from *N* =  27 short post-mortem interval (PMI) control tissues; data derived from [50,51,55,58,61,65,66,67,68,69]; this group included 9 males and 18 females, mean age 75.2  ±  9.5 years; all post-mortem intervals (PMIs; death to brain freezing at −81 °C) were less than 3.3 h (see Table 1); each of these miRNAs yielded high (≥3000) units of signal strength on LC Sciences miRNA analytical arrays Genechip (proprietary MRA-1001-miRNA microfluidic chip analytical platform; 2650 small non-coding RNAs (sncRNAs) were analyzed; LC Sciences Corporation, Houston, TX, USA; https://lcsciences.com/company/technology/technology-microarray/(accessed on 26 December 2022) and ranked as the 54 most abundant miRNAs detected in the superior temporal lobe neocortex; except for miRNAs listed here, all other miRNAs are at a significantly lower abundance; note that there is a relatively high basal abundance of the let-7a to let-7i group of miRNAs; also note that AD-abundant miRNAs such as miRNA-9, miRNA-30b, miRNA-34a, miRNA-125b, miRNA-146a, and miRNA-155 exhibit high relative variability in mean abundance in control brains and are significantly up-regulated in AD brains; in human neuronal-glial (HNG) cells in primary culture, most of these same miRNAs are induced by the pro-inflammatory transcription factor NF-kB (p50/p65) [66]; these same miRNAs are up-regulated in AD brains, especially as AD progresses; superior temporal lobe-enriched miRNAs overlain in gray (and their mRNA targets) have been strongly implicated in AD [23,50,51,52,53,54,55,57,66]; miRNAs with a single asterisk such as miRNA-7, miRNA-9, miRNA-30b, miRNA-34a, miRNA-125b, and miRNA-146a are moderately abundant in control of the adult human superior temporal lobe neocortex (Brodmann Area 22); many of these same miRNAs are implicated in other infectious, inflammatory, and/or immunological diseases [11,12,23,24,25,26,27,28,29,63,65,66]; miRNAs with two asterisks are among the most abundant miRNAs in this same region of the human temporal lobe neocortex (see manuscript text); a superscript of ‘f’ denotes miRNAs known to be induced by the pro-inflammatory transcription factor NF-kB (p50/p65) complex [65,66].

## Data Availability

All data used in this review are openly available and freely accessible on MedLine (www.ncbi.nlm.nih.gov (accessed on 26 December 2022) where they are listed by the last names of the individual authors.

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
