# Peer review of "Endogenous miRNA-Based Innate-Immunity against SARS-CoV-2 Invasion of the Brain"

_ijms, 2023, doi:10.3390/ijms24043363_

Round 1
Reviewer 1 Report
Please find attached the comments.

Reviewer 2 Report
The manuscript entitled “Endogenous miRNA-based innate-immunity against SARS-CoV-2 invasion of the brain” by Lukiw and Pogue is very interesting in the field.
I have a few suggestions/comments for the authors
1. The main aim of this review is to focus on the human brain, miRNAs, and SARS-CoV-2 invasion. Authors have considered describing AD, PrD and age-related neurodegeneration and the miRNAs related to them. I would like to get more neurodegenerative disorders like Parkinson, Huntington, multiple system atrophy, etc. and their related miRNAs with specific diseases for a better understanding of the review.
2. In section 4, the authors described ‘Human Biochemical Individuality’ and I think it requires a more elaborative description on a scientific basis including genome-wide association studies, precision medicine and other more advanced recent technical tools, considering not only ‘biochemical’ but also ‘genetical’. The current manuscript describes the genetic level of the SARS-CoV-2 infection in the human brain and the area should be more focused.
3. Table-1: hsa-miRNA-30b, -146a, -155, -7a, -23a, -29a, -125b are associated with several other diseases like inflammatory diseases, infections, and immunological diseases. There are not very specific miRNAs for only brain tissue. I think it would be worth mentioning in the manuscript all these details. Why some of these miRNAs are highlighted in grey?
4. Section 5 describes the ‘unanswered question’ that is very intriguing, but some questions like (i), is a basic question in molecular biology and how this is relevant to the current manuscript?
The question (iii), none of the cited articles suggest the interaction among all human miRNAs and human SARS-CoV-2 ssvRNA. Could you please provide more information on how all human miRNAs interact with SARS-CoV-2 ssvRNA?
Could you please elaborate the question (iv), how could multiple miRNAs bind to the same target site? Is it possible to bind to the same target site by multiple miRNAs? I think this binding depends on the entropy of the miRNA-mRNA target binding.
I feel that questions (v) and (vi) are very much hypothetical since no real analysis or data is available for SARS-CoV-2, but question (vi) is relevant with proper data/analysis results.
For all questions other than (viii) and (ix), how those are relevant to the current manuscript when the focus is on the human brain? Also, question (viii) is focused on the other forms of RNAs when the focus of the current manuscript is on the miRNAs.
Round 2
Reviewer 2 Report
With the revised version and as mentioned by the authors in the cover letter, as a small review, the current version is suitable for publication in the journal.